# PTTG1/ZEB1 Axis Regulates E-Cadherin Expression in Human Seminoma

**DOI:** 10.3390/cancers14194876

**Published:** 2022-10-05

**Authors:** Emanuela Teveroni, Fiorella Di Nicuolo, Edoardo Vergani, Giada Bianchetti, Carmine Bruno, Giuseppe Maulucci, Marco De Spirito, Tonia Cenci, Francesco Pierconti, Gaetano Gulino, Pierfrancesco Bassi, Alfredo Pontecorvi, Domenico Milardi, Francesca Mancini

**Affiliations:** 1International Scientific Institute Paul VI, Fondazione Policlinico Universitario A. Gemelli IRCCS, 00168 Rome, Italy; 2Division of Endocrinology, Fondazione Policlinico Universitario A. Gemelli IRCCS, 00168 Rome, Italy; 3Department of Neuroscience, Section of Biophysics, Università Cattolica del Sacro Cuore, 00168 Rome, Italy; 4Fondazione Policlinico Universitario A. Gemelli IRCCS, 00168 Rome, Italy; 5Division of Anatomic Pathology and Histology, School of Medicine, Università Cattolica del Sacro Cuore, 00168 Rome, Italy; 6Department of Urology, Fondazione Policlinico Universitario A. Gemelli IRCCS, 00168 Rome, Italy

**Keywords:** TGCTs, seminoma, PTTG1, ZEB1, E-Cadherin, EMT, invasiveness

## Abstract

**Simple Summary:**

Seminoma represents one of the most common neoplasms in Caucasian males between 15 and 40 years old. The molecular pathways underlying its clinical behavior are far from being understood yet. We previously demonstrated that nuclear Pituitary-tumor transforming-gene 1 (PTTG1), overexpressed in several neoplasms, promotes invasiveness through its transcriptional target matrix-metalloproteinase-2 (MMP2). PTTG1 sustains the migratory and invasive properties of cancer cells through the induction of the epithelial-to-mesenchymal transition (EMT). E-Cadherin (E-CAD) repression is the first step of EMT. Therefore, we investigated the role of PTTG1 in EMT in human seminoma using an in vitro and in vivo model and through Atlas database interrogation. Our data showed a PTTG1-mediated E-CAD transcriptional repression through Zinc finger E-box binding homeobox 1 (ZEB1), a master regulator of the EMT process. Our data provide insights into the molecular characterization of seminoma, promoting PTTG1 as a prognostic marker useful in human seminoma clinical management.

**Abstract:**

(1) Background: PTTG1 sustains the EMT process and the invasiveness of several neoplasms. We previously showed the role of nuclear PTTG1 in promoting invasiveness, through its transcriptional target MMP2, in seminoma in vitro models. Here, we investigated the key players involved in PTTG1-mediated EMT in human seminoma. (2) Methods: Two seminoma cell lines and four human seminoma tumor specimens were used. E-Cadherin gene regulation was investigated using Western blot, real-time PCR, and luciferase assay. Immunoprecipitation, ChIP, RE-ChIP, and confocal microscopy analysis were performed to evaluate the interplay between PTTG1 and ZEB1. Matrigel invasion and spheroid formation assays were applied to functionally investigate PTTG1 involvement in the EMT of seminoma cell lines. RNA depletion and overexpression experiments were performed to verify the role of PTTG1/ZEB1 in E-Cadherin repression and seminoma invasiveness. E-Cadherin and ZEB1 levels were analyzed in human testicular tumors from the Atlas database. (3) Results: PTTG1 transcriptionally represses E-Cadherin in seminoma cell lines through ZEB1. The cooperation of PTTG1 with ZEB1 has a significant impact on cell growth/invasion properties involving the EMT process. Analysis of the Atlas database of testicular tumors showed significantly lower E-Cadherin levels in seminoma, where PTTG1 showed nuclear staining. Finally, PTTG1 and ZEB1 strongly localize together in the periphery of the tumors. (4) Conclusions: These results strengthen the evidence for a role of PTTG1 in the EMT process in human seminomas through its cooperation with the transcriptional repressor ZEB1 on the E-Cadherin gene. Our data enrich the molecular characterization of seminoma, suggesting that PTTG1 is a prognostic factor in seminoma clinical management.

## 1. Introduction

Testicular germ cell tumors (TGCTs) account for 1% of all cancers in men, representing the most frequent neoplasm for 20 to 40 year olds. Of these, seminomas are the most common subtype [1]. Germ cell neoplasia in situ (GCNIS), the pre-invasive stage of a TGCT (equivalent to “carcinoma in situ”) has the potential to evolve into nonseminoma or seminoma cancer [2,3].

The majority of TGCT patients show clinical stage I disease for both nonseminomatous and seminomatous histology. In stage I seminoma patients, the prognosis is very good, with an overall survival rate of 98%. However, about 15–20% of patients develop tumor recurrence [4,5]. The primary tumor size and rete testis invasion (RTI) are associated with this phenomenon and are used in clinical practice to guide decision making on adjuvant treatment [6]. In particular, RTI is an unfavorable prognostic factor and recent studies have focused on the identification of specific RTI biomarkers [7]. This is in line with the lack of efficacious prognostic factors for seminoma management and for disease progression in terms of surrounding tissue invasion.

The pituitary tumor-transforming gene 1 (PTTG1) is a securin and its overexpression has been reported in several neoplasms, frequently those of endocrine origin [8,9,10,11,12]. PTTG1 has transcriptional activity and exerts its function either by directly binding to DNA or by interacting with other proteins such as other transcription factors [13]. PTTG1 overexpression leads to altered sister chromatid segregation during cell division, causing aneuploidy and genetic instability [14]. Its contribution to tumorigenesis is also associated with the promotion of invasiveness through its transcriptional activity. Indeed, PTTG1 contributes to the metastatic process and tumor progression through the transactivation of many targets, such as matrix metalloproteinase 2 (MMP-2) [15]. In seminoma, PTTG1 oncogenic properties were strictly linked with its nuclear localization. Indeed, we previously demonstrated that, in the periphery of seminoma tumors, PTTG1 is localized in the nucleus, suggesting that the more invasion-prone cancer area requires nuclear localization of the securin [16]. In addition, the nuclear expression of PTTG1 has been correlated to a more aggressive phenotype, and to tumor recurrence [17,18]. Moreover, PTTG-binding factor (PBF) (a PTTG1-interacting partner) which promotes the shift of the securin into the nucleus [19], has been found to be involved in the progression of different tumors [20,21,22]. Lastly, we have recently demonstrated the role of PTTG1 nuclear localization in promoting invasiveness and metastatic process in different seminoma cell lines via the increase of MMP2 levels [23]. The Atlas database analysis confirmed the prevalent nuclear localization of PTTG1 in seminoma tumors, and the strong correlation between its nuclear localization and the increased expression of MMP2 versus nonseminoma tumors [23]. Thus, it is presumable that the increased nuclear expression of PTTG1 could represent an eligible prognostic factor in seminomas.

It has been demonstrated that PTTG1 sustains the migratory and invasive properties of cancer cells through the induction of the epithelial-to-mesenchymal transition (EMT) [24]. In an ovarian tumor cell line, it was reported that PTTG1 induces EMT by downregulation of E-Cadherin (CDH1 gene), the master regulator of this process [25]. EMT is an evolutionarily conserved process that is tightly regulated by the interplay among a complex web of signaling pathways that leads to the activation of a network of EMT-inducing transcription factors (EMT-TF). This reversible transdifferentiation process involves a remodeling of the cytoskeleton and a loss of cell-to-cell adhesion. One of the earliest steps is the loss of E-Cadherin (hereafter E-CAD) function, a key determinant of adherens junctions, through the direct repression of its promoter [26]. ZEB1 and ZEB2 proteins belong to the zinc finger family of transcription factors. They have two flanking zinc finger clusters that bind to paired CACCT(G) E-box-like promoter elements present on the CDH1 promoter [27]. ZEB1 and ZEB2 present several protein-binding domains that include the Smad-, CtBP-, and p300-P/CAF-interaction domains, which are instrumental in the control of their transcriptional activity. Therefore, ZEB factors act as transcriptional repressors through their interaction with the CtBP corepressor [28]. They can activate transcription through their interaction with coactivators such as p300 and P/CAF [28]. A dysregulated expression of ZEB1 has been reported in many human tumors, where it is generally believed to be responsible for cell migration, invasion, and metastasis [28]. The aim of the present work was to investigate the role of PTTG1 in seminoma cancer progression, and particularly its role in the regulation of the EMT process. We evaluated the ability of PTTG1 to transcriptionally suppress the E-CAD protein in seminoma cell lines. This phenomenon is dependent on the presence of ZEB1. Importantly, we found that PTTG1 directly interacts with ZEB1 and that both proteins are present on the endogenous CDH1 promoter, demonstrated by Re-ChIP experiments, suggesting that PTTG1 acts as a co-repressor of ZEB1 on this promoter. The interplay between PTTG1 and ZEB1 was confirmed by functional assays involving the EMT process, namely the invasion assay and the sphere-forming ability assay. It is of note that analysis of the Atlas database of testicular tumors [29] supported the in vivo role of the PTTG1/ZEB1/E-CAD axis in human seminoma. The interrogation of this database revealed that in seminoma tumors, where PTTG1 is more localized in the nucleus than in the nonseminoma subtype, a significantly lower E-CAD expression is observed compared to the observations of nonseminomas. This phenomenon is not coupled with an increase in ZEB1 levels in seminoma, supporting the hypothesis that nuclear PTTG1 cooperates with ZEB1-dependent E-CAD repression in human seminoma tumors. In addition, using immunoprecipitation, we demonstrated the interaction of PTTG1 and ZEB1 in human seminoma specimens from patients that underwent therapeutic orchidectomy for seminoma. Importantly, confocal immunofluorescence analysis of one of these specimens revealed that PTTG1 and ZEB1 significantly colocalized in specific isolated cells present at the periphery of the tumors where PTTG1 showed nuclear staining, as we previously reported [23]. Taken together, these data support the role of PTTG1/ZEB1 interplay in driving E-CAD repression in vivo, and hence in promoting EMT in specific areas of human seminoma tumors.

## 2. Materials and Methods

### 2.1. Cell Culture and Transfections

SEM-1 and JKT-1 seminoma cell lines (kindly provided by Dr. A. L. Epstein) and 293T were cultured in RPMI supplemented with 10% FBS (Millipore, Burlington, MA, USA) and stable glutamine (glutamax, Thermofisher, Waltham, MA, USA). Transient transfections were performed using Jet Prime Polyplus according to the manufacturer’s instructions (Polyplus, Illkirch-Graffenstaden, France). Plasmids used were: pcDNA3.1, pCMV (as control vectors, CTR), pcDNA3.1-PTTG1, pCMV-FLAG-PTTG1, pCIneo HA-PBF, pGL3-ECAD-LUC (pGL3-Luc carrying −178/+92 E-CAD proximal promoter kindly provided by Dr. Antonio García de Herreros), pCIneo-ZEB1 (kindly provided by Dr. Giulia Fontemaggi), and shCTL e sh-ZEB1 (pGIPZ-shRNAmir vectors, kindly provided by Dr Thomas Brabletz). PTTG1 siRNA (siPTTG1) and control siRNA (siCTL) were purchased from Thermofisher (Stealth RNAi). Cells were transfected using RNAiMAX reagent according to the manufacturer’s instructions (Thermofisher).

### 2.2. Immunofluorescence

Cells were fixed with 3.7% formaldehyde for 15 min at RT, permeabilized with 0.05% Triton X-100 in PBS, and blocked with 5% bovine serum albumin (BSA). Cells were incubated with primary antibodies (rabbit α-PTTG1, Abcam; mouse α-ZEB1, Santa Cruz) and then incubated with goat Alexa Fluo-488 anti-rabbit IgG and/or goat Alexa Fluo-594 anti-mouse IgG (Molecular Probes, Eugene, OR, USA).

### 2.3. Confocal Microscopy and Colocalization Analysis

Cells were imaged with an inverted confocal microscope (Nikon A1-MP). Fluorescence images were collected on three separated channels (excitation: 402 nm, emission: 450/50 nm for the blue channel; excitation: 488 nm, emission: 525/50 nm for the green channel; excitation: 561 nm, emission: 595/50 nm for the red channel) using a 60× oil-immersion objective with 1.4 Numerical Aperture (NA). Internal photon multiplier tubes collected 1024 × 1024 pixel images in 16 bit at 0.063 ms dwell time.

The colocalization of PTTG1 and ZEB1 at the nuclear level was evaluated using the Colocalization Threshold plugin available in the open-source software ImageJ (NIH). Fluorescence colocalization was graphically represented in scatterplots in which the intensity of green channel is reported versus the intensity of the red channel for each pixel, as reported in [30]. PTTG1-ZEB1 colocalization was quantitatively evaluated through the Mander’s colocalization coefficient *tM_2_* [23], defined as follows:$$tM_{2}=\frac{\sum_{i}R_{i,coloc}}{\sum_{i}R_{i}}\;\mathrm{where}\;\left\{\begin{array}{c} R_{i,\;coloc}=R_{i}\;\mathrm{if}\;G_{i}>0\\ R_{i,\;coloc}=0\;\mathrm{if}\;G_{i}=0 \end{array}\right.$$
with *R* and *G* denoting two probes and *tM*_2_ providing the fraction of *R*, i.e., the fraction of pixels expressing ZEB1, in compartments expressing *G*, which is PTTG1. The threshold value was automatically identified by applying the Costes method [31], and the coefficient evaluated only for pixels above the threshold.

### 2.4. Western Blot Analysis and Immunoprecipitation

For immunoprecipitation (IP), cells or frozen tissues were lysed in IP lysis buffer (50 mM Tris–Cl, pH 7.5, 150 mM NaCl, 1% Nonidet P-40, 1 mM EDTA) containing a mix of protease inhibitors (Boehringer, Ingelheim am Rhein, Germany). For IP, lysates were incubated overnight with the indicated antibody, under gentle rocking at 4 °C. The immunocomplexes were then isolated using Dynabeads according to the manufacturer’s instructions (ThermoFisher, Waltham, MA, USA). In-vitro-translated proteins used for immunoprecipitation were produced using a TNT-coupled wheat germ extract system (Promega, Medison, WI, USA). For Western blot analysis, cells were lysed in RIPA buffer (50 mM Tris–Cl, pH 7.5, 150 mM NaCl, 1% Nonidet P-40, 0.5% Na deoxycholate, 0.1% SDS, 1 mM EDTA) containing a mix of protease inhibitors (Boehringer). Proteins were separated by SDS–PAGE and then transferred onto PVDF membranes (Millipore, Burlington, MA, USA). Membranes were developed using enhanced chemiluminescence (ECL westar, Cynagen, Bologna, Italy). Bands were analyzed using a chemiluminescence imaging system, Alliance 2.7 (UVITEC, Cambridge, UK) and quantified by the software Alliance V_1607. The following primary antibody were used: rabbit α-PTTG1 (Abcam, Cambridge, UK), rabbit α-PTTG1 (ThermoFisher), mouse α-ZEB1 (Santa Cruz, Santa Cruz, CA, USA), rabbit α-ZEB1 (Genetex, Irvine, CA, USA), rabbit α-ECAD (Genetex), mouse α-HA (BioLegend, San Diego, CA, USA), mouse α-actin monoclonal antibody C-40 (Sigma, St. Louis, MO, USA).

### 2.5. RNA Preparation and Quantitative Reverse-Transcription PCR

Total RNA extraction was performed with TRIzol reagent (Invitrogen, Waltham, MA, USA). For quantitative reverse-transcription PCR (Q-RTPCR), total RNA (1 μg) was reverse transcribed using the Gene Amp kit (Applied Biosystems, Waltham, MA, USA) and subjected to PCR amplification using SYBR Green PCR Master Mix (Applied Biosystems). The following primers were used: CDH1 5′-CTGGGACTCCACCTACAGAAAGTT-3′ and 5′-CCAGAAACGGAGGCCTGAT-3′; β-ACTIN 5′-CCAACCGCGAGAAGATGAC-3′ and 5′-TAGCACAGCCTGGATAGCAA-3′.

### 2.6. Luciferase Assay

In the luciferase reporter assay, cells were transfected with pGL3-ECAD-LUC and pCMV-Renilla (ratio 10:1), which was used as an internal control, along with the different combinations of plasmids (reported in the text). At 2 days after transfection, firefly and Renilla luciferase activities were measured using the Dual Luciferase Reporter Assay System (Promega) with Varioscan Lux (ThermoScientific), and the Renilla luciferase activity relative to the firefly luciferase activity was calculated.

### 2.7. Chromatin Immunoprecipitation

For the ChIP of PTTG1 and ZEB1, 1.5 × 10^6^ cells were grown in complete medium for 24 h. ChIP was performed according to Raspaglio et al. [32]. Briefly, chromatin was crosslinked with 1% formaldehyde (Sigma-Aldrich) and then glycine was added to a final concentration of 0.125 M. After 5 min, cells were washed, scraped with PBS, and centrifuged. Pellet samples were then resuspended in hypotonic buffer (10 mM Hepes/KOH pH 7.5, 10 mM EDTA, 0.5 mM EGTA, 0.25% Triton X-100) on ice for 15 min to isolate nuclei. After washing, the nucleus pellets were resuspended in lysis buffer (50 mM Tris/HCl pH 8, 10 mM EDTA, 0.5% SDS plus protease inhibitors) and sonication was performed using an ultrasonic processor VCX130. Sonicated lysate was diluted 1:5 in ChIP dilution buffer (150 mM NaCl, 2 mM EDTA, 20 mM Tris/HCl pH 8, 1% Triton X-100). Two percent of the not- diluted lysate was kept for input control. Diluted lysates were precleared with the addition of protein G Protein Magnetic Beads (Invitrogen, Waltham, MA, USA) (previously blocked with salmon sperm and bovine serum albumin) for 1 h at 4 °C. Subsequently, 20 µg of chromatin extract was immunoprecipitated overnight on a rotating platform at 4 °C with the following antibodies: PTTG1 (Abcam) and ZEB1 (Genetex). No antibody was used as a negative control. Immunoprecipitated chromatin was conjugated with G Protein Magnetic Beads (Invitrogen). After extensive washing, bound DNA fragments were eluted and analyzed via qPCR of CDH1 promoters, using SYBR Green Master Mix (biorad, Hercules, CA, USA).

For the Re-ChIP experiments, after ZEB1 immunoprecipitation, immune complexes were eluted in Re-ChIP elution buffer (TE, 2% SDS, 15 mM DTT, supplemented with protease inhibitor). Chromatin was then diluted 1:20 in ChIP dilution buffer and subjected to the second IP α-PTTG1 antibody. Promoter occupancy was assessed by PCR amplification of the −86/+60 sequence of the human E-CAD promoter, which contains ZEB1-binding sites (E-Boxes). The following primers were used, as described in Vandewalle et al. [33]: 5′ GGCCGGCAGGTGAAC 3′ (forward), 5′ GGGCTGGAGTCTGAACTGAC 3′ (reverse).

### 2.8. Human Seminoma Samples

The study was conducted in accordance with the guidelines of the Declaration of Helsinki. All patients gave written, informed consent to anonymous use of their data for research purposes, and the protocol was approved by the ethics committee of the Fondazione Policlinico Universitario “A. Gemelli”, Rome, Italy (protocol ID 4824). Testicular tissues of patients who underwent an orchidectomy for seminoma were collected at the Department of Surgical Pathology Fondazione Policlinico Universitario “A. Gemelli”. Formalin-fixed and paraffin-embedded tissues were used for confocal microscopy analysis, whereas tissues immediately frozen at −80 °C after surgery were used for immunoprecipitation assays. To perform confocal analysis, deparaffinized tissue slides were rehydrated using a graded alcohol solution. Antigen retrieval was performed in 10 mM citrate buffer at pH 6.0 for 10 min in a microwave oven. Slides were then washed in distilled water and sequentially rinsed in PBS. All primary antibodies were incubated overnight at 4 °C. Sections were then incubated at room temperature with primary antibody rabbit monoclonal PTTG1 (Securin, clone EPR3240, Abcam, Cambridge, UK, 1:500 for 1 h). The PTTG1 was visualized using the Alexa Fluor 488-conjugated goat anti-Rabbit IgG secondary antibody (ThermoFisher Scietific, USA, 1:1000 for 1 h). After, the slides were rinsed in PBS and then incubated with mouse monoclonal ZEB1 (Santa Cruz, 1:50 for 1 h). These antibodies were visualized using the Alexa Flour 594-conjugated goat anti-Mouse IgG secondary antibody (ThermoFisher Scientific, 1:1000 for 1 h). Slides were rinsed in PBS and mounted in Vectashield (H-1000, Vector Laboratories, Peterborough, UK), and double immunofluorescence slides were examined under confocal immunofluorescence microscopy. Table 1 shows the main histopathological features of patients and seminomas.

### 2.9. Invasion Assay

Invasion assays were performed as previously described [19]. Briefly, the cells (1.5 × 104 cells in serum-free medium) were seeded in the upper well of a Transwell chamber (8 mm pore size) precoated with 10 mg/mL growth-factor-reduced Matrigel (BD Biosciences, Franklin Lakes, NJ, USA). The lower well was filled with complete growth medium as a chemoattractant. After incubation for 2 days at 37 °C, noninvaded cells on the upper surface of the filter were removed with a cotton swab, and migrated cells on the lower surface of the filter were fixed and stained. Cells were imaged using phase-contrast microscopy (Leica Microsystems, Wetzlar, Germany, magnification 10×). Invasiveness was determined by counting cells in five microscopic fields per well, and the extent of invasion was expressed as an average number of cells per microscopic field.

### 2.10. Sphere-Formation Assay

For spheroid formation, 2500 cells/well were seeded in a 96-well ultra-low-attachment (ULA) multiplate, diluted in 100 μL of medium in the presence or absence of 20% methylcellulose (MC) to promote intercellular aggregation. Cells were imaged using phase-contrast microscopy after 5 days (Leica Microsystems, magnification 10×).

### 2.11. Statistical Analysis

Results were expressed as values of mean ± standard deviation (SD). The statistical test used was paired two-tailed student *t*-test. A *p* < 0.05 was considered as significant. The Atlas database was evaluated via unpaired *t*-test with Welch’s correction (that is, not assuming equal SDs). The software used was GraphPad Prism 7.04 (San Diego, CA, USA). In particular, in the Atlas database, we analyzed mRNA levels of E-CAD and ZEB1 in nonseminoma (N-S; N = 65) and seminoma (S; N = 68) specimens. Data were expressed in FPKM (fragments per kilobase of exon model per million reads mapped), a unit used to estimate gene expression based on RNA-seq data.

## 3. Results

### 3.1. Analysis of PTTG1 Regulation of E-Cadherin in Seminoma Cell Lines

We previously demonstrated the involvement of PTTG1 in the migration/invasion properties of seminoma cell lines. Since the EMT process is closely related to tumor cell migration and invasion [34], we wondered if, in seminomas, the securin is involved in this process. To this end, we modulated PTTG1 protein levels in JKT-1 and SEM-1 seminoma cell lines, which show a more aggressive phenotype among the seminoma lines, and we evaluated the expression of E-CAD, since it is known that its decreased expression is one of the earliest steps in the EMT process.

Initially, we overexpressed increasing doses of PTTG1 and analyzed E-CAD protein levels in SEM-1 cells (Figure 1A). Immunoblot analysis of whole-cell extracts showed that as PTTG1 levels increased, E-CAD levels significantly decreased (Figure 1A). The E-CAD decreasing trend upon PTTG1 overexpression was investigated using a luciferase reporter construct carrying −178/+92 E-Cadherin proximal promoter (E-CAD LUC) [35]. Figure 1B shows that an increasing amount of PTTG1 protein caused a gradual decrease of luciferase signal, suggesting that PTTG1 regulates E-CAD at the transcriptional level. To further demonstrate the repression of E-CAD by PTTG1, we depleted its expression in JKT-1 cells using small interference RNA (siRNA). The RNA interference of PTTG1 significantly upregulated E-CAD expression at both protein level and in terms of promoter activity (Figure 1C and D, respectively). Additionally, the decreased in PTTG1 level caused an increase in the endogenous mRNA levels of E-CAD in JKT-1 cells (Figure 1E).

To assess the role of PTTG1 nuclear localization in the observed phenomenon, we overexpressed PBF, which has been reported to mediate PTTG1 nuclear translocation [19], together with PTTG1 in SEM-1 cells (Figure 1 F,G). Western blot analysis revealed that PBF overexpression significantly increased the repressive function of PTTG1 on E-CAD expression at the levels of both protein and promoter activity (Figure 1F and G, respectively) suggesting, as conceivable, the need for the nuclear localization of PTTG1 to observe the above-mentioned activity. Taken together, these data indicate that PTTG1 is able to transcriptionally repress E-CAD in seminoma cell lines.

### 3.2. PTTG1 Repression of E-Cadherin Depends on ZEB1

Since ZEB1 is a key element of a pool of transcription factors that control EMT [28] and represses E-CAD by binding to E box elements in its promoter region [35,36], we wondered if the effect of PTTG1 on E-CAD transcription is dependent on ZEB1 activity. To address this point, we analyzed E-CAD protein expression upon PTTG1 overexpression together with a simultaneous reduction of ZEB1 protein levels through short hairpin RNA of ZEB1 (shZEB1) in SEM-1 cells. Immunoblot analysis showed that PTTG1-induced repression of E-CAD was almost abrogated upon shZEB1 exposure (Figure 2A). This result was confirmed at the transcriptional level using both the E-CAD LUC reporter construct and the analysis of the endogenous mRNA levels of E-CAD (Figure 2B,C, respectively). Vice versa, PTTG1 significantly repressed E-CAD to a greater extent in the presence of ZEB1 overexpression in the luciferase assay (Appendix A). Taken together, these results suggest that PTTG1 represses E-CAD transcription via the ZEB1 protein.

To further investigate the crosstalk between PTTG1 and ZEB1, we questioned whether the two proteins were able to physically interact with each other. To this end, we first performed immunoprecipitation analysis upon overexpression of FLAG-tagged PTTG1 (FLAG-PTTG1) in an independent system, namely the 293T cell line. As shown in Figure 3A, these two proteins specifically interacted. In addition, binding of endogenous proteins was observed in both JKT-1 and SEM-1 seminoma cell lines (Figure 3B and Appendix A). Importantly, the PTTG1/ZEB1 interaction was direct, as demonstrated by the reciprocal immunoprecipitations of the in vitro translated proteins (Figure 3C,D).

These data prompted us to investigate whether the PTTG1/ZEB1 interaction occurs at the chromatin level by chromatin immunoprecipitation analysis (ChIP). At first, we analyzed the presence of PTTG1, or ZEB1 as a positive control, on the endogenous E-CAD promoter in the SEM-1 cell line (Figure 3E). ChIP experiments revealed that PTTG1 and ZEB1 are present on the E-CAD promoter (Figure 3E). The specificity of PTTG1 occupancy on the E-CAD promoter was validated via ChIP experiment under RNA interference (siRNA) of PTTG1 (Appendix A). In order to investigate whether the presence of PTTG1 on the E-CAD promoter is dependent on ZEB1, we performed the anti-PTTG1 ChIP experiment by decreasing the expression of ZEB1 via shRNA. As shown in Figure 3F, the lowering of ZEB1 levels caused a decrease in the presence of PTTG1 on the E-CAD promoter, suggesting that ZEB1 is crucial for PTTG1 localization on this promoter. To definitively confirm the PTTG1/ZEB1 simultaneous presence at the chromatin level and to confirm the action of PTTG1 as co-repressor of ZEB1 on E-CAD promoter, we performed a Re-ChIP experiment. We performed the ChIP anti-PTTG1 after the immunoprecipitation of the chromatin with anti-ZEB1 antibody and demonstrated simultaneous occupancy of the E-CAD promoter by PTTG1 and ZEB1 (Figure 3G).

Overall, these experiments show that PTTG1 and ZEB1 directly interact and that this complex is simultaneously present on the E-CAD promoter.

### 3.3. Analysis of PTTG1/ZEB1 Interplay in Seminoma Cell Lines

To find out whether the observed relationship between PTTG1 and ZEB1 has functional consequences, we performed two commonly used assays to study the EMT pathway: the Matrigel invasion assay and the sphere-forming ability assay.

To this end, we made use of a Matrigel-coated transwell chamber assay. We measured the invasion rate of SEM-1 cells upon PTTG1 overexpression in the presence or absence of ZEB1. As reported in Figure 4A,B, PTTG1 overexpression caused a significant increase in the invasion rate of SEM-1 cells that was almost completely abolished by the shRNA of ZEB1. Importantly, such behavior was also confirmed in the sphere-forming assay (Figure 4C,D). Indeed, the increase of the sphere-forming ability caused by PTTG1 overexpression was rescued by the decrease of ZEB1 expression by shRNA (Figure 4C,D). These data indicate that the cooperation of PTTG1 with ZEB1 has a significant impact on cell growth/invasion properties involving the EMT process.

To determine whether the role of PTTG1 observed in cell lines has a translational impact in human seminomas, we wondered whether the PTTG1/ZEB1/E-CAD axis was confirmed in human testicular tumors. Previously, we demonstrated that PTTG1 has a predominant nuclear localization in seminomas compared to nonseminomas. This feature was correlated with an increase of MMP2 levels [23], known to be upregulated by the same PTTG1. In this regard, we wondered whether E-CAD shows differential expression in the two testicular tumor subclasses. Taking advantage of the Atlas database [29] we analyzed E-CAD mRNA levels in nonseminoma (N-S; N = 65) and seminoma (S; N = 68) specimens of TGCTs. Interestingly, the analysis showed a highly significant lower level of E-CAD in seminoma samples compared to nonseminoma samples (Figure 5A), supporting the role of nuclear PTTG1 in driving E-CAD transcriptional repression in human seminomas. To verify that the observed phenomenon relies on PTTG1, we analyzed ZEB1 levels in seminomas, since an augmented expression in this subclass could belittle PTTG1 role. As shown in Figure 5B, ZEB1 levels were significantly lower in seminomas than in nonseminomas, supporting our hypothesis that PTTG1’s cooperation with ZEB1 is crucial in the transcriptional repression of E-CAD in human seminomas.

To understand in depth the interplay between PTTG1 and ZEB1 in human seminomas, we performed immunoprecipitation and confocal microscopy analysis of human seminoma specimens from patients (Table 1: P1–4) that underwent therapeutic orchidectomy for seminomas (Figure 5C,D). Immunoprecipitation of PTTG1 performed in samples derived from three frozen seminoma specimens (S1, S2, S3), obtained from the above-mentioned patients (P1–3), revealed that PTTG1 interacts specifically with ZEB1 in all samples analyzed, albeit to varying degrees, reflecting the inherent variability of human tumor tissues (Figure 5C). The reciprocal immunoprecipitation of ZEB1 in S1 and S2 confirmed the previous results (Appendix A). Importantly, as shown in Figure 5D, confocal immunofluorescence analysis of a human seminoma specimen (S4 derived from patient P4) showed that PTTG1 and ZEB1 significantly colocalized in isolated cells present on the fringes of the tumors where PTTG1 showed nuclear staining, as was previously reported [16]. In the panel, the fluorescence emission of PTTG1 and ZEB1 are shown separately in green and red, respectively. Looking at the overlap of the two signals, represented on the right (Merge), a marked colocalization, resulting in yellow-colored pixels, can be observed at the nuclear level. This colocalization was quantitatively evaluated through Mander’s *tM_2_* coefficient, for which the mean value (96.2 ± 2.2%) highlighted a strong overlap in the two components. Taken together, these results strengthen the role of PTTG1 in the EMT process in human seminomas, focusing its function on the cooperation with the transcriptional repressor activity of ZEB1, specifically on the E-CAD gene.

## 4. Discussion

The findings of this paper detail the molecular pathways involved in PTTG1-mediated tumor progression in human seminoma, strengthening understanding of its role in the EMT process.

We previously reported that PTTG1 expression, specifically its nuclear localization in seminoma cell lines is responsible for a more aggressive phenotype via its migratory properties, Matrigel invasion capability and MMP-2 proteolytic activity [23].

Furthermore, we confirmed the role of nuclear PTTG1 in human seminoma taking advantage of the Atlas database of TGCTs [29]. The Atlas database interrogation revealed that in seminomas, there was an increased nuclear PTTG1 localization along with higher levels of MMP-2 compared to nonseminoma tumors. Taken together, these data support the hypothesis that nuclear PTTG1 can promote tumor progression, driving MMP-2 levels, and indicates that the subcellular localization of this securin represents a specific feature of seminoma among TGCTs, suggesting its possible use as a prognostic factor in clinical practice. Extracellular matrix degradation through MMP-2, a fundamental step in the acquisition of metastatic properties of tumor cells, takes part in the wider and more complex process of EMT. This prompted us to deepen our understanding of the molecular mechanism underlying the role of PTTG1 in seminoma tumor progression, with a special focus on EMT process.

The EMT is a series of molecular events through which epithelial cells lose many of their epithelial characteristics and acquire a mesenchymal phenotype [37]. It is well known that EMT has a key role in embryogenesis [26] and in multiple physiological and pathological conditions, such as fibrogenesis and tumor progression [38,39]. One of the hallmarks of EMT is the downregulation of E-CAD [40]. In fact, the loss of adherens junctions and the consequent reorganization of the cytoskeleton has been demonstrated to be an early event that defines the invasion-prone phenotype [39].

Shah and Kakar demonstrated that PTTG1 is able to induce the EMT process by regulation of multiple key players in ovarian cancer cell lines [25]. Specifically, they found that PTTG1 mediated upregulation of EMT inducers Twist, Snail, and Slug, along with downregulation of E-CAD, through expression and secretion of TGF-β [25]. Furthermore, it was reported that PTTG1 induced EMT in breast cancer cells [24] by promoting AKT phosphorylation, which in turn, mediated Snail increase (a well-known E-CAD transcriptional repressor [41]).

Here, we demonstrated, for the first time in seminoma cells, an E-CAD decrease upon dose-dependent PTTG1 overexpression. Furthermore, we found that the PTTG1-mediated E-CAD downregulation occurs at the transcriptional level. This finding suggests and confirms that PTTG1, a well-known transcriptional activator, can act as a transcriptional repressor too. Accordingly, Chesnokova and collaborators reported that PTTG1 overexpression suppressed p21 promoter activity in CHO cells, in a dose-dependent manner [42]. To confirm the PTTG1-mediated E-CAD regulation, we depleted PTTG1 levels by siRNA in JKT-1 cells, which show higher PTTG1 amounts in comparison to SEM-1, as we formerly reported [23]. We found that as PTTG1 decreased, the expression of E-CAD increased through enhanced transcriptional activity on its promoter. To confirm the PTTG1-mediated transcriptional repression of E-CAD occurring in the nucleus, we overexpressed low levels of PTTG1 with or without PBF, previously reported as a mediator of PTTG1 nuclear translocation [19]. As expected, PBF overexpression significantly increased the PTTG1 repression of E-CAD. Overall, these data support and detail the pivotal role of PTTG1 nuclear localization as a marker of its detrimental activity in seminoma cells.

The EMT process is marked by a deep transcriptional reprogramming of the cell, requiring a specific repressive molecular platform in which many players are involved. Among them, ZEB1 is a dominant transcription factor that promotes EMT at both the transcriptional and translational levels [43].

A crosstalk has been reported between ZEB1 and PTTG1, involving miR-3666 in cervical cancer cell lines [44]. Li and colleagues, indeed, showed that PTTG1 overexpression inhibited miR-3666, a transcriptional repressor of ZEB1. This in turn led to an increase in ZEB1 levels, driving the cellular gain of invasive properties [44].

In the present study, we reported for the first time a direct molecular interaction between PTTG1 and ZEB1 with the function of a transcriptional repression complex. We found that the PTTG1-mediated E-CAD transcriptional inhibition is strongly impaired by the depletion of ZEB1, suggesting the need for ZEB1 in this repressive function of the securin.

To further detail the molecular mechanism behind PTTG1/ZEB1 cooperation, we carried out immunoprecipitation in both an unrelated cellular model, 293-T, and in a seminoma cell line. We first found the interaction between the two proteins in all cellular systems tested. We then showed that this interaction is direct by performing immunoprecipitation with in vitro translated proteins. Moreover, using ChIP and RE-ChIP experiments, we demonstrated that PTTG1 and ZEB1 act simultaneously on chromatin, specifically on the E-CAD promoter. Taken together, these results show for the first time a new co-repressional platform in seminoma cells, indicating that PTTG1/ZEB1 cooperation is essential for PTTG1 to exert its E-CAD regulation.

In functional assays, we further demonstrated that PTTG1/ZEB1 interaction mediates the securin oncogenic properties in seminoma cell lines. Indeed, Matrigel cell invasion assays showed that the PTTG1-mediated invasiveness was almost abrogated against a ZEB1-depleted background. We confirmed these data by performing spheroid-formation assays, supporting the hypothesis that PTTG1/ZEB1 collaboration is needed for PTTG1 to carry out its effect on tumor progression, specifically through EMT-promoting activity.

To verify the clinical impact of these findings, we made use of the Atlas database’s human testicular cancer searches [29]. The use of this database showed that E-CAD mRNA levels are significantly lower in seminoma tumors in comparison to nonseminoma tumors. This result fits with our previous data about the specific nuclear localization of the securin in seminomas, confirming that this subcellular localization characterizes PTTG1 tumor-promoting activity through repression of E-CAD. Moreover, the Atlas database analysis of ZEB1 levels revealed that its mRNA levels are decreased in seminoma in comparison to nonseminoma tumors; this result indicates that the decreased levels of E-CAD in human seminoma is not entirely due to ZEB1, underscoring the role of PTTG1 in E-CAD transcriptional suppression. Finally, to further confirm the PTTG1/ZEB1 interplay from a clinical point of view, we analyzed the molecular interaction via immunoprecipitation and confocal microscopy of human seminoma specimens obtained from patients who underwent therapeutic orchidectomy. In these tumor samples, we confirmed the binding between the two proteins and, importantly, confocal immunofluorescence analysis revealed that this interaction occurs in isolated cells at the peripheral area of the tumor. All together, these data support the pivotal role of nuclear PTTG1 in driving the invasion-prone cell population of human seminoma, and shed a light on the underlying mechanism in which the interplay with ZEB1 is crucial. Therefore, the results of this research represent a further step in understanding the partners and crosstalk of PTTG1 that contribute to its oncogenic activity and represent potential prognostic factors in human seminoma.

## 5. Conclusions

The present study is an extension of our previous research which showed higher PTTG1 nuclear localization in the invasion-prone cell population at the periphery of human seminoma. This phenomenon has been correlated with increased MMP-2 expression and with more invasive properties of seminoma cell lines. Here, we focused on the role of PTTG1 in the EMT process in seminoma. We found that PTTG1 was able to transcriptionally repress the E-CAD gene depending on ZEB1 interaction. Analysis of TGCT’s Atlas database strongly supported our results, showing significantly lower levels of E-CAD in seminomas compared to nonseminomas. Importantly, we demonstrated the physical interaction of PTTG1 and ZEB1 in human seminoma specimens. These data support our hypothesis that PTTG1’s cooperation with ZEB1 is crucial in the transcriptional repression of E-CAD in human seminomas. Overall, the results of this research represent a further step in understanding the partners and interactions of PTTG1 that contribute to its oncogenic activity and represent potential prognostic factors in human seminomas.

## Figures and Tables

**Figure 1 cancers-14-04876-f001:**
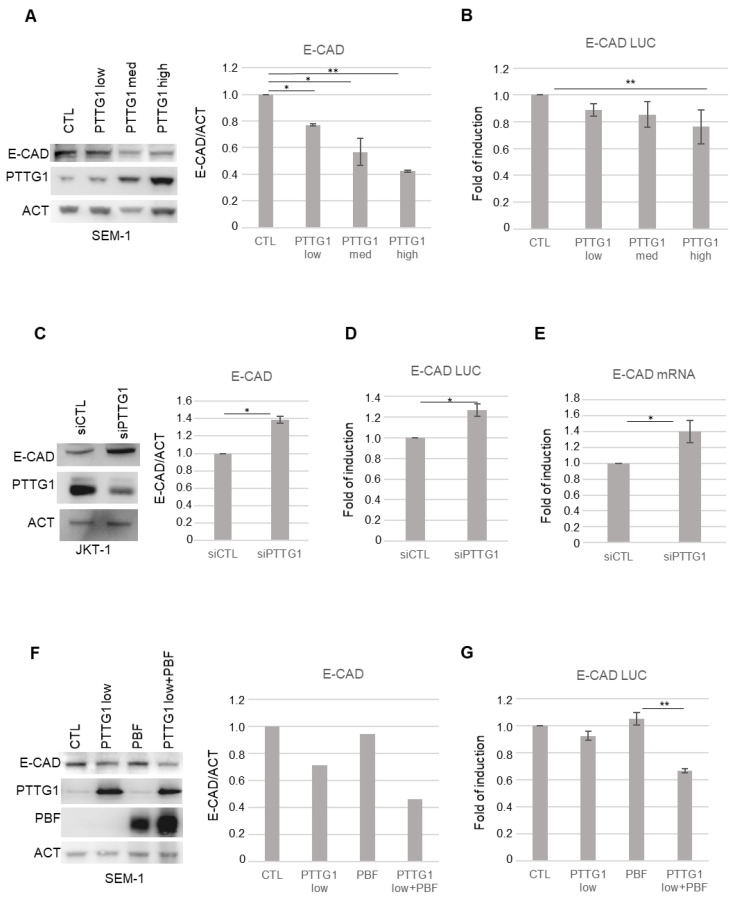
PTTG1 regulation of E-CAD in seminoma cell lines. (**A**) Left panel: representative Western blot analysis (Wb) of the indicated proteins in SEM-1 cell line. “Low”, “Med”, and “High” indicate increasing amount of transfected PTTG1 plasmid. Right panel: Histogram shows the ratio of densitometric values of E-CAD to actin (ACT) referred to left panel. Mean ± SD of three independent biological replicates is shown (N = 3; * = *p* < 0.05, ** = *p* < 0.01, two-tailed unpaired *t*-test). (**B**) Histogram shows the fold of induction of luciferase activity, normalized to Renilla internal control signal, set arbitrarily to 1 in control transfection (CTL). Mean ± SD of three independent biological replicates is shown (N = 3; ** = *p* < 0.01, two-tailed unpaired *t*-test). (**C**) Left panel: representative Western blot analysis (Wb) of the indicated proteins in JKT-1 cell line upon siRNA of PTTG1 (siPTTG1) or control (siCTL). Right panel: Histogram shows the ratio of densitometric values of E-CAD to actin (ACT) referred to left panel. Mean ± SD of three independent biological replicates is shown (N = 3; * = *p* < 0.05, two-tailed unpaired *t*-test). (**D**) Histogram shows the fold of induction of luciferase activity, normalized to Renilla signal, set arbitrarily to 1 in control transfection (CTL). Mean ± SD of three independent biological replicates is shown (N = 3; * = *p* < 0.05, two-tailed unpaired *t*-test). (**E**) Histogram shows the fold of induction of E-CAD mRNA relative to actin upon siRNA of PTTG1 (siPTTG1) and control (siCTL), set arbitrarily to 1 in control transfection (CTL). Mean ± SD of three independent biological replicates is shown (N = 3; * = *p* < 0.05, two-tailed unpaired *t*-test). (**F**) Left panel: representative Western blot analysis (Wb) of the indicated proteins in SEM-1 transfected with control plasmid (CTL), low dose of PTTG1 plasmid (PTTG1 low), PBF plasmid (PBF), and PTTG1 low plus PBF. Right panel: Histogram shows the ratio of densitometric values of E-CAD to actin (ACT) referred to left panel. Mean ± SD of two independent biological replicates is shown. (**G**) Histogram shows the fold of induction of luciferase activity in SEM-1 transfected as in (**F**), normalized to Renilla signal, set arbitrarily to 1 in control transfection (CTL). Mean ± SD of three independent biological replicates is shown (N = 3; ** = *p* < 0.01, two-tailed unpaired *t*-test).

**Figure 2 cancers-14-04876-f002:**
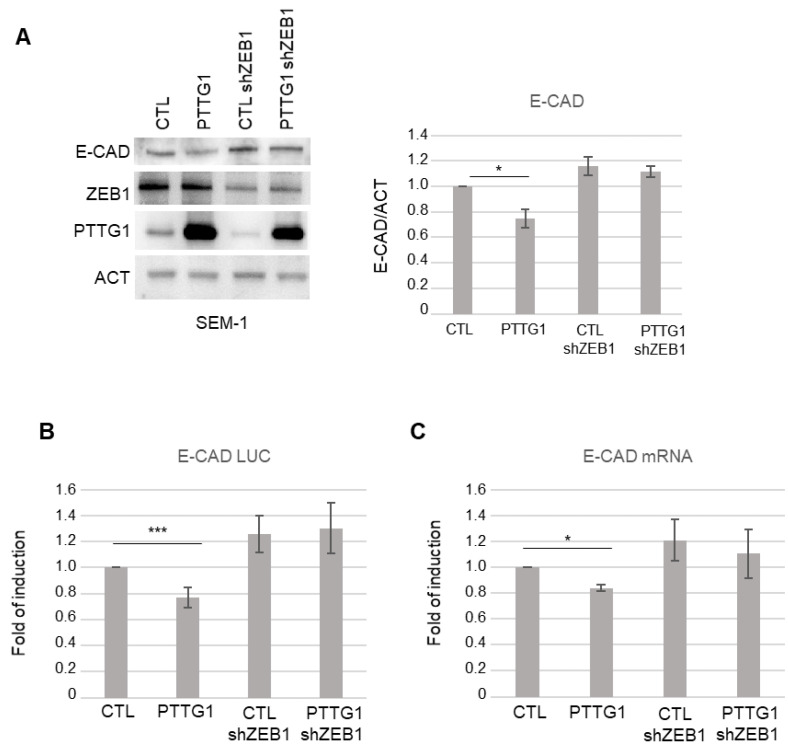
PTTG1 regulation of E-CAD depends on ZEB1 presence. (**A**) Left panel: representative Western blot analysis (Wb) of the indicated proteins in SEM-1 cell line upon PTTG1 overexpression and shRNA of ZEB1 (shZEB1) or control (CTL). Right panel: Histogram shows the ratio of densitometric values of E-CAD to actin (ACT). Mean ± SD of three independent biological replicates is shown (N = 3; * = *p* < 0.05). (**B**) Histogram shows the fold of induction of luciferase activity in the indicated transfections (as in (**A**)), normalized to Renilla signal, set arbitrarily to 1 in control transfection (CTL). Mean ± SD of three independent biological replicates is shown (N = 3; *** = *p* < 0.001, two-tailed unpaired *t*-test). (**C**) Histogram shows the fold of induction of E-CAD mRNA relative to actin in the indicated transfections (as in (**A**)). Control transfection (CTL) is set arbitrarily to 1. Mean ± SD of three independent biological replicates is shown (N = 3; * = *p* < 0.05, two-tailed unpaired *t*-test).

**Figure 3 cancers-14-04876-f003:**
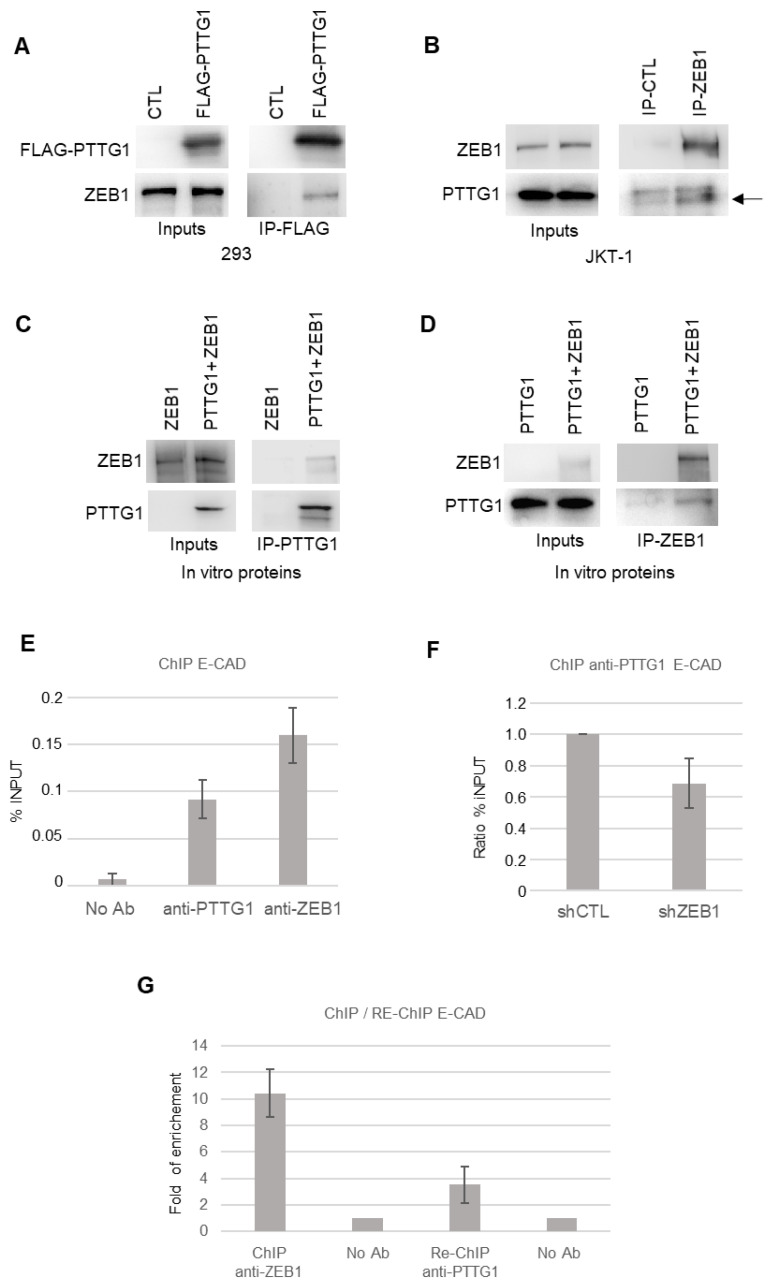
PTTG1 interacts with ZEB1. (**A**) Western blot analysis (Wb) of the indicated proteins. Inputs represent 1/10 of the total protein used in immunoprecipitation of FLAG-PTTG1 (IP-FLAG) in 293T cells transfected with FLAG-PTTG1 or control plasmid (CTL). (**B**) Western blot analysis (Wb) of the indicated proteins. Inputs represent 1/10 of the total protein used in immunoprecipitation of control (IP-CTL) or ZEB1 (IP-ZEB1) in JKT-1 cells. Arrow indicates PTTG1-specific band in right panel. (**C**) Western blot analysis (Wb) of the indicated proteins. Inputs represent 1/10 of the total protein used in immunoprecipitation of PTTG1 (IP-PTTG1) using ZEB1 alone or PTTG1 and ZEB1 in vitro translated proteins. (**D**) Western blot analysis (Wb) of the indicated proteins. Inputs represent 1/10 of the total protein used in immunoprecipitation of ZEB1 (IP-ZEB1) using PTTG1 alone or PTTG1 and ZEB1 in vitro translated proteins. (**E**) Chromatin immunoprecipitation analysis with the indicated antibodies (No Ab; anti-PTTG1; anti-ZEB1) on E-CAD promoter in SEM-1 cells. Histogram shows the % of input chromatin used for the immunoprecipitation. (**F**) Chromatin immunoprecipitation analysis of PTTG1 protein on E-CAD promoter upon shRNA of ZEB1 (shZEB1) or control (shCTL). Histogram shows the ratio of % of input chromatin used for the immunoprecipitation. Control ratio is set arbitrarily to 1. (**G**) Sequential chromatin immunoprecipitation (Re-ChIP) analysis of PTTG1 protein on E-CAD promoter after ChIP of ZEB1. Histogram shows fold of enrichment on E-CAD promoter upon ChIP of ZEB1 (ChIP anti-ZEB1) and upon sequential Re-ChIP of PTTG1 (Re-ChIP anti-PTTG1). Inputs are set arbitrarily to 1.

**Figure 4 cancers-14-04876-f004:**
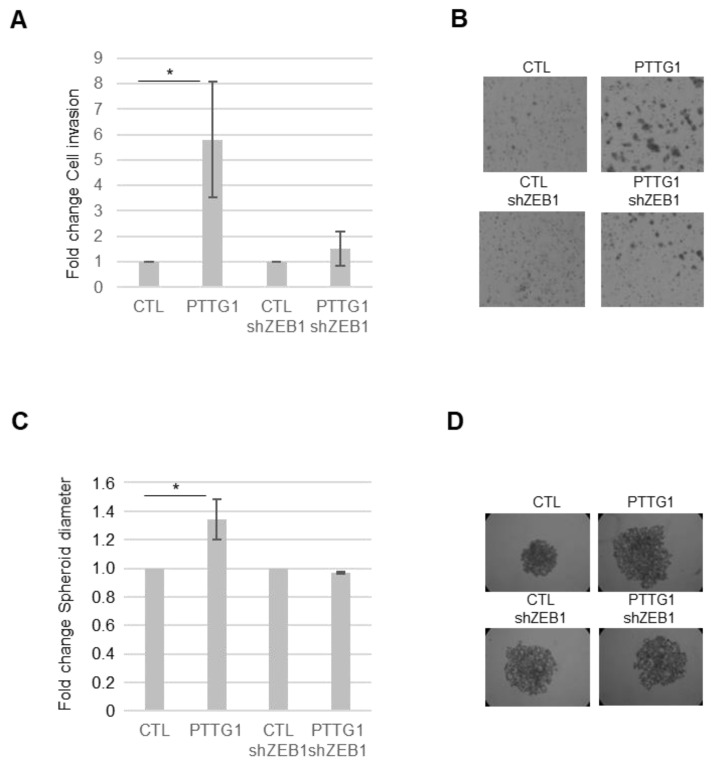
PTTG1/ZEB1 regulates invasion of a seminoma cell line. (**A**) Histogram shows Matrigel cell invasion of SEM-1 cells upon the indicated transfections. The number of invaded cells in both control samples (CTL and CTL-shZEB1) is arbitrarily set to 1 (N = 3, * = *p* < 0.05). (**B**) Representative images of Matrigel cell invasion assay reported in (**A**). (**C**) Histogram shows changes in spheroid cell diameter in SEM-1 cells upon the indicated transfections. The spheroid cell diameter in both control samples (CTL and CTL-shZEB1) is arbitrarily set to 1 (N = 3, * = *p* < 0.05). (**D**) Representative pictures of spheroid cell formation assay reported in (**C**).

**Figure 5 cancers-14-04876-f005:**
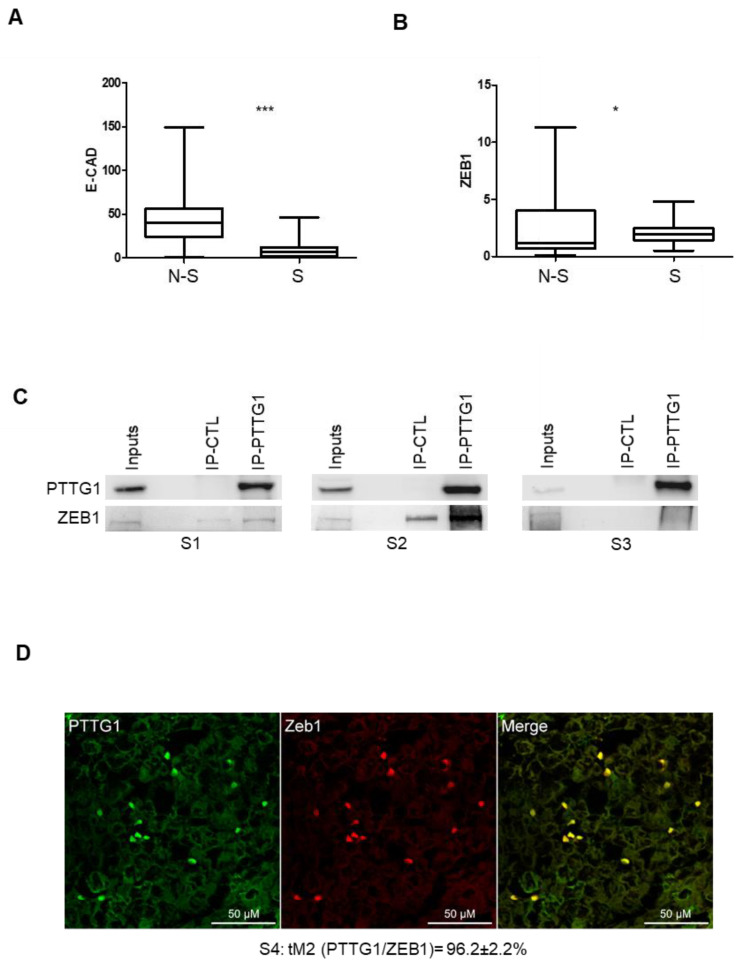
PTTG1/ZEB1 interplay in human seminoma specimens. (**A**) Box plot of mRNA levels of E-CAD in nonseminoma (N-S; N = 65) and seminoma (S; N = 68) specimens in Atlas database (https://www.proteinatlas.org/ENSG00000039068-CDH1/pathology/testis+cancer (accessed on 3 November 2021) (*** = *p* < 0.001)). (**B**) Box plot of mRNA levels of ZEB1 in nonseminoma (N-S; N = 65) and seminoma (S; N = 68) specimens in Atlas database (https://www.proteinatlas.org/ENSG00000148516-ZEB1/pathology/testis+cancer (accessed on 3 November 2021). (* = *p* < 0.05)). (**C**) Western blot analysis (Wb) of the indicated proteins. Inputs represent 1/20 of the total protein used in immunoprecipitation of control (IP-CTL) or PTTG1 (IP-PTTG1) in three different human seminoma specimens (S1, S2, S3). (**D**) Representative pictures of merged (yellow signal) confocal immunofluorescence analysis of PTTG1 (green signal) and ZEB1 (red signal) in a human seminoma specimen (S4). Quantification of the nuclear colocalization of PTTG1 and ZEB1, evaluated through the Mander’s *t_2_* (*tM_2_*) coefficient and expressed as percentage is reported below. Scale bar is 50 μM.

**Table 1 cancers-14-04876-t001:** Histopathological features of the analyzed samples of human testicular seminomas.

	P1	P2	P3	P4
**Age (years)**	23	27	44	54
**History of cryptorchidism**	no	no	yes	no
**Tumor Size (mm) and Site**	35 mm, right testis	40 mm, right testis	8 mm, right testis	90 mm, left testis
**pTNM**	pT1pNx	pT2pNx	pT1pNx	pT2pNx
**Risk Factors**	-	Rete testis+ Vascular invasion+	Rete testis+	Vascular invasion+ Linfoadenopathies+

## Data Availability

The data presented in this study are available in this article (and Appendix A).

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
