# Peer review of "PTTG1/ZEB1 Axis Regulates E-Cadherin Expression in Human Seminoma"

_cancers, 2022, doi:10.3390/cancers14194876_

Round 1

Reviewer 1 Report

The paper by Teveroni et al. discloses the role of PTTG1 and ZEB1 in regulating E-CAD. Moreover, it examines their role in seminoma cancer progression, particularly in the epithelial to mesenchymal transition process. The present manuscript extends previous work of the research group. In my opinion, manuscript may be suitable for publication in Cancers, once all the comments are satisfactory addressed and the revised manuscript is improved.

Specific comments:

1. Line 279: PTTG1 regulates E-CAD at transcriptional level only partially. There seem to be another (even more important than transcription) PTTG1-mediated mechanism responsible for regulation of E-CAD.

2. Why different cell lines were used to examine PTTG1-mediated E-CAD regulation. Some aspects were addressed using SEM-1, the other by JKT-1. What was the reason for that?

3. The authors should provide the direct evidence that nuclear localization of PTTG1 is prerequisite for its repressive function on E-CAD expression.

4. Why 293T cell line was used for immunoprecipitation experiments (Figure 3a)? Was there any particular reason for not using seminoma cell lines for this purpose? Figure 3b is rather confusing in showing that even IP-CTL brings down PTTG1.

5. The authors conclude that there is interplay/cooperation between PTTG1 and ZEB1. Such an interpretation leads one to believe that these two proteins contribute about equally to processes that they regulate. However, ZEB1 clearly regulates PTTG1, and acts upstream of PTTG1 in the process, based on data presented in Figure 2a, Figure 3f, Figure 4a et cetera. The authors suggest that PTTG1 is a prognostic factor in seminoma clinical management. Would ZEB1 be even better candidate for this?

6. Would the PTTG1/ZEB1/E-CAD axis be differently expressed in seminomas with and without rete testis invasion? Could the PTTG1/ZEB1/E-CAD axis be used to stratify clinical stage I seminoma patients in terms of their further management after inguinal radical orchiectomy, adjuvant chemotherapy or active surveillance (watchful waiting)? Can authors add data to their manuscript, based on the The Cancer Genome Atlas dataset, correlating the level of expression of the PTTG1/ZEB1/E-CAD axis with clinico-pathological characteristics of clinical stage I seminoma patients, namely progression-free and overal survivals?

7. Atlas database analysis should be descibed in more detail, particularly methodology.

8. The second (ZEB1-independent) mechanism responsible for recruitment of PTTG1 to E-CAD promoter may exist, based on data presented in Figure 3F. Do the authors have any idea what it could be?

9. Is there difference in terms of invasivity/metastatic spread between seminomas and non-seminomas? If so, could it be attribued to the PTTG1/ZEB1/E-CAD axis?

10. It should be indicated which specimen belongs to which patient. Or does it mean that S1 belongs to P1, S2 belongs to P2 and so on? What was the reason to use different speciment in confocal microscopy studies compared to immunoprecipitation studies?

11. Figures should be better described. Figures should be able to stand on their own, so that the reader can easily understand them. For example Figure 3, part presenting data obtained using in vitro translated proteins is very hardly understandable.

12. Manuscript requires further editing, despite careful English editing by native English person, as stated in Acknowledgements. There are some grammatical and typing errors, e.g. line 20 … securin - :s overexpression …, line 277-278 … Figure 1b show …

Author Response

Response to Reviewer 1 Comments

  1. Line 279: PTTG1 regulates E-CAD at transcriptional level only partially. There seem to be another (even more important than transcription) PTTG1-mediated mechanism responsible for regulation of E-CAD.

Response 1: Thank you for your comment; indeed, the levels of the E-CAD protein following overexpression of PTTG1 decrease largely than the luciferase signal, relative to transcriptional activity on its promoter. However, the results obtained with both assays were statistically significant. The observed phenomenon is probably due to the different sensitivity of the used assays (Western Blot and Luciferase assay). Moreover, the transfection for Luciferase assay involves the use of more plasmids (PTTG1, E-CAD-LUC, Renilla) than in Western blot assay (PTTG1), that could interfere with the final amount of PTTG1 overexpression and consequently of E-CAD levels. Anyway, we cannot exclude an additional post-transcriptional effect of PTTG1 on E-CAD.

  1. Why different cell lines were used to examine PTTG1-mediated E-CAD regulation. Some aspects were addressed using SEM-1, the other by JKT-1. What was the reason for that?

Response 2: In the present study, we used the more aggressive seminoma cell lines used in our previous work about seminoma (doi:10.3390/CANCERS13020212). In particular, we chose to use the SEM-1 cell line for the overexpression experiments, as it possesses intermediate levels of the PTTG1 protein, while we used the JKT1 cell line for the siRNA experiments, as it possesses the highest levels of PTTG1.

  1. The authors should provide the direct evidence that nuclear localization of PTTG1 is prerequisite for its repressive function on E-CAD expression.

Response 3: Thank you for your comment; We have addressed this point with two different approaches.

  1. It has been reported that PBF mediates PTTG1 nuclear translocation (doi:10.1074/jbc.M910105199) and we previously confirmed this PBF activity on PTTG1 in the aggressive seminoma cell line JKT1 (doi:10.3390/CANCERS13020212). Taking advantage of PBF protein, we showed that its overexpression leads to an increase of PTTG1 dependent E-CAD transcriptional repression indicating a strong link between this phenomenon to PTTG1 nuclear localization.
  2. Using ChIP and ReChIP experiments, we demonstrate the presence of PTTG1 on chromatin and specifically its occupancy on E-CAD promoter together with ZEB1 protein.

  1. Why 293T cell line was used for immunoprecipitation experiments (Figure 3a)? Was there any particular reason for not using seminoma cell lines for this purpose? Figure 3b is rather confusing in showing that even IP-CTL brings down PTTG1.

Response 4: We used 293T cell line as seminoma unrelated system to perform Immunoprecipitation experiment in FLAG-PTTG1 overexpression condition, as a first approach to evaluate the possible PTTG1/ZEB1 interaction. Then we confirmed the PTTG1/ZEB1 binding in two seminoma cell lines using endogenous proteins.

Thank you for your comment about Figure 3b. In the mentioned panel, there is an upper not specific band present in both IP-CTL and IP-ZEB1. We have revised the figure adding an arrow that indicates PTTG1 specific band.

  1. The authors conclude that there is interplay/cooperation between PTTG1 and ZEB1. Such an interpretation leads one to believe that these two proteins contribute about equally to processes that they regulate. However, ZEB1 clearly regulates PTTG1, and acts upstream of PTTG1 in the process, based on data presented in Figure 2a, Figure 3f, Figure 4a et cetera. The authors suggest that PTTG1 is a prognostic factor in seminoma clinical management. Would ZEB1 be even better candidate for this?

Response 5: Thank you for your comment; in the present research we didn’t assume that PTTG1 and ZEB1 contribute equally to the observed processes but we conclude that there is an interplay between PTTG1 and ZEB1 in terms of PTTG1 co-repressional activity with the repressor ZEB1 on E-CAD promoter. Indeed, ZEB1 is one the main repressor of E-CAD that binds to paired CACCT(G) E-box-like promoter elements while PTTG1 binds ZEB1 regulating its repressive activity (acting as a ZEB1 co-repressor).

We suggest that nuclear PTTG1 is a prognostic factor in seminoma based on its activity on MMP2 (previously reported in doi:10.3390/CANCERS13020212) and on its the cooperation with ZEB1. Moreover, from Atlas database we showed that PTTG1 is localized in the nucleus specifically in seminoma where it probably carries out its oncogenic activities. On the other hand, ZEB1 levels are lower in Seminoma vs Non-seminoma, indicating that ZEB1 itself is not a good candidate as prognostic factor in seminoma clinical management whereas it could be the co-localization between the two proteins in a subset of invasive prone cells (as we showed in figure 5d).

  1. Would the PTTG1/ZEB1/E-CAD axis be differently expressed in seminomas with and without rete testis invasion? Could the PTTG1/ZEB1/E-CAD axis be used to stratify clinical stage I seminoma patients in terms of their further management after inguinal radical orchiectomy, adjuvant chemotherapy or active surveillance (watchful waiting)? Can authors add data to their manuscript, based on the The Cancer Genome Atlas dataset, correlating the level of expression of the PTTG1/ZEB1/E-CAD axis with clinico-pathological characteristics of clinical stage I seminoma patients, namely progression-free and overal survivals?

Response 6: Thank you for your suggestions. Unfortunately, the rete testis invasion (RTI) was not reported in Atlas database. The study of the possible involvment of PTTG1/ZEB1/E-CAD axis with RTI could be very interesting because RTI is an unfavourable prognostic factor correlated with tumour recurrence. We missed to mention RTI in the paper and we added this information in the introduction session of the revised manuscript.

As you pointed out, we stratified clinical stages I of seminomas from Atlas with PTTG1/ZEB1/E-CAD axis correlating the proteins levels with the different Stage I subgroups (1, 1a, 1b and 1s) but we did not find any significant correlation (Response Figure 1, see below). The analysis was performed using one-way analysis of variance (ANOVA) followed by a post-hoc test (Bonferroni test). We did not add this analysis in the manuscript because, regarding the focus of our study, it would be better to compare Stage I with Stage II and III, in which cancer has spread outside the testis. For now, we cannot perform this analysis because there are only few seminoma Stage II and III (n=5) in Atlas database.

Finally, as you suggested, we correlates the level of expression of the PTTG1/ZEB1/E-CAD axis with the progression-free of stage I seminoma patients present in Atlas database. We did not found a significant correlation (Response Figure 2, see below). The analysis was performed using Linear regression test. As we stated previously, we did not add this analysis in the manuscript because it would be better to compare clinico-pathological characteristics of Stage I with Stage II and III.

  1. Atlas database analysis should be described in more detail, particularly methodology.

Response 7: Thank you for your suggestions; we improve the methodology of Atlas database analysis in Materials and Methods section.

  1. The second (ZEB1-independent) mechanism responsible for recruitment of PTTG1 to E-CAD promoter may exist, based on data presented in Figure 3F. Do the authors have any idea what it could be?

Response 8: Thank you for your comment; in Figure 3F we showed that the lowering of ZEB1 levels by shRNA causes a decrease in the presence of PTTG1 on E-CAD promoter, suggesting that ZEB1 is crucial for PTTG1 localization on such promoter. The observed reduction of PTTG1 on E-CAD promoter was not very strong because, unfortunately, the decrease of ZEB1 levels upon shRNA was not complete (because of the low efficiency of the shRNA technique; see Fig. 2a).

  1. Is there difference in terms of invasivity/metastatic spread between seminomas and non-seminomas? If so, could it be attribued to the PTTG1/ZEB1/E-CAD axis?

Response 9: Thank you for your comment; from Atlas database the local recurrence is 10% for seminomas and 20% for non-seminomas; moreover, the % of distant metastasis are 1.5 and 7.5 for seminomas and non-seminomas respectively. These differences are not statistically significant and we cannot correlate them to the PTTG1/ZEB1/E-CAD axis.

  1. It should be indicated which specimen belongs to which patient. Or does it mean that S1 belongs to P1, S2 belongs to P2 and so on? What was the reason to use different specimen in confocal microscopy studies compared to immunoprecipitation studies?

Response 10: Thank you for your suggestions; your observation is correct, the seminoma specimens S1-4 belong to patients P1-4. We have revised the manuscript to clarify this issue. We use S1-3 for immunoprecipitation analysis because such tissues were frozen suitable for perform cellular lysates, while S4 was a paraffin-embedded tissue suitable for Immunofluorescence analysis.

  1. Figures should be better described. Figures should be able to stand on their own, so that the reader can easily understand them. For example Figure 3, part presenting data obtained using in vitro translated proteins is very hardly understandable.

Response 11: Thank you for your suggestions; your observation is correct, Figure 3 legend was not very comprehensible. We modified it. Moreover, we revised all figure legends.

  1. Manuscript requires further editing, despite careful English editing by native English person, as stated in Acknowledgements. There are some grammatical and typing errors, e.g. line 20 … securin - :s overexpression …, line 277-278 … Figure 1b show …

Response 11: Thank you for your suggestions; we revised the manuscript.                

Please find details in the attachment.

Reviewer 2 Report

The paper of Teveroni et al., entitled:” PTTG1/ZEB1 axis regulates E-Cadherin expression in human seminoma” well describe the molecular pathways involved in PTTG1-mediated tumor invasion in human seminoma and its involvement in the Epithelial Mesenchymal Transition (EMT) process. I appreciated the description of the physical interaction of PTTG1 and ZEB1 in human seminoma samples. I think that this paper is very interesting and highlight the importance of PTTG1/ZEB1 axis in human seminoma. However, I have a minor revision for the authors. Below are my comments, as I believe that the authors will be able to better explain some questions.

1.     The authors did not report the data obtained on both cell lines in the figures. For example, in the figure 1a, 1f, 2a is represented only one cell line (SEM-1). Did the authors also obtain the same result on JKT-1 cell line? Why wasn’t it shown? Have the experiments been carried out on both cell lines? I ask the authors to show the images of both cell lines, if available. If not, please the authors to argue the reason why this data have not been reported.

2.     Have luciferase assays been performed on both cell lines? The histograms shown in the figures are representative for only one cell line. Please the authors to explain this point or, if possible, report data from both cell lines in the text and figures.

3.     In the figure 1f, page 8/22, the immunoblotting image shows a bubble in “PTTG1 low” lane. Please the authors to replace this image with another one. 

4.     I ask the authors to revise the text of all figures legends. Some are unclear especially those of the supplementary. In addition, in the figure legend 1c there is an error. The authors wrote:” C Left panel: representative Western blot analysis (Wb) of the indicated proteins in SEM-1 cell line upon siRNA of PTTG1 (siPTTG1) and control (siCTL).”, (lane 302-303, page 9/22) but under of the immunoblotting image is indicated JKT-1. Please, the authors to modify the text.

5.     In figure 3c and 3d, the authors reported under immunoblotting “in vitro proteins”. Which cell line are they referring to? Please the authors to specify better in the imagine and in the figure legend. 

6.     The Matrigel assay are carry out only in SEM-1? Please the authors to argue this point.

7.     The authors sent the original immunoblot images. Please, the authors to mark the molecular weight of the marker for each image.

Author Response

Response to Reviewer 2 Comments

  1. The authors did not report the data obtained on both cell lines in the figures. For example, in the figure 1a, 1f, 2a is represented only one cell line (SEM-1). Did the authors also obtain the same result on JKT-1 cell line? Why wasn’t it shown? Have the experiments been carried out on both cell lines? I ask the authors to show the images of both cell lines, if available. If not, please the authors to argue the reason why this data have not been reported.

Response 1: Thank you for your comment; in the present study, we used the more aggressive seminoma cell lines carrying different amount of PTTG1 protein used in our previous work (doi:10.3390/CANCERS13020212). In particular, we chose to use only the SEM-1 cell line for all the overexpression experiments, as it possesses intermediate levels of the PTTG1 protein, while we used the JKT1 cell line for the siRNA experiments, as it possesses the highest levels of PTTG1.

  1. Have luciferase assays been performed on both cell lines? The histograms shown in the figures are representative for only one cell line. Please the authors to explain this point or, if possible, report data from both cell lines in the text and figures.

Response 2: Thank you for your comment; we made use of both SEM1 and JKT1 cell lines for luciferase assays. In particular, we used SEM1 cells in Figure 1b, 1g and 2b, while we used JKT1 in Figure 1d.

  1. In the figure 1f, page 8/22, the immunoblotting image shows a bubble in “PTTG1 low” lane. Please the authors to replace this image with another one. 

Response 3: Thank you for your suggestions; your observation is correct, we replaced it with a better image.

  1. I ask the authors to revise the text of all figures legends. Some are unclear especially those of the supplementary. In addition, in the figure legend 1c there is an error. The authors wrote:” C Left panel: representative Western blot analysis (Wb) of the indicated proteins in SEM-1 cell lineupon siRNA of PTTG1 (siPTTG1) and control (siCTL).”, (lane 302-303, page 9/22) but under of the immunoblotting image is indicated JKT-1. Please, the authors to modify the text.

Response 4: Thank you for your suggestions; your observation is correct, we improved some figure legends and in particular, we corrected the mistake in cell line in Figure 1c legend (replacing SEM-1 with the right cells JKT-1).

  1. In figure 3c and 3d, the authors reported under immunoblotting “in vitro proteins”. Which cell line are they referring to? Please the authors to specify better in the imagine and in the figure legend. 

Response 5: In vitro-translated proteins used for immunoprecipitation in Figure 3c and d, were produced by TNT-coupled wheat germ extract system (Promega) that is a wheat germ extract in which are present the transcription and translation machineries and where  is possible to translate proteins encoded by a DNA plasmid. Therefore, it is an in vitro acellular system that we used to demonstrate that the PTTG1/ZEB1 interaction is direct and not mediated by other proteins.

  1. The Matrigel assay are carry out only in SEM-1? Please the authors to argue this point.

Response 6: For all overexpression experiments, we chose to use only the SEM-1 cell line because it possesses intermediate levels of the PTTG1 protein. In particular, in Matrigel assay, we aimed to demonstrate, in functional assays, the ZEB1 dependency of overexpressed PTTG1 activity on E-CAD (as in figure 2 with molecular assays).

  1. The authors sent the original immunoblot images. Please, the authors to mark the molecular weight of the marker for each image.

Response 7: Thank you for your suggestions; we added the molecular weight of the marker for each immunoblot image.

Reviewer 3 Report

The paper on “PTTG1/ZEB1 axis regulates E-Cadherin expression in human Seminoma” is a research article reporting new and interesting data on the molecular characterization of seminoma, considering PTTG1 as a useful prognostic marker in the clinical management of human seminoma.

The manuscript is well written, and the experimental conditions have been described.

Could the authors clarify whether seminoma patients have a clinical history of cryoprtochidism?

Author Response

Response to Reviewer 3 Comments

Point 1: Could the authors clarify whether seminoma patients have a clinical history of cryoprtochidism?

Response 1: Thank you for your comment. About patients who underwent an orchidectomy for seminoma in our Hospital, we add this information in Table 1.

With regard to patients from Atlas Database (doi:10.1016/J.CELREP.2018.05.039), they reported a total TGCTs personal history of cryptorchidism of 17%. In particular, they showed 25% in seminomas and 7.7% in non-seminomas tumours.
